# Exploring data quality and use of the routine health information system in Ethiopia: a mixed-methods study

The Operational Research and Coaching for Analysts (ORCA)- participants & team, Abyot Adane,[1] Tewabe M Adege,[2] Mesoud M Ahmed,[2] Habtamu A Anteneh [ID],[2] Emiamrew S Ayalew [ID],[2] Della Berhanu [ID],[3] Netsanet Berhanu,[2] Misrak Getnet,[4] Tesfahun Bishaw,[2] Joanna Busza,[3] Eshetu Cherinet,[2] Mamo Dereje,[2] Tsega H Desta,[2] Abera Dibabe,[2] Heven S Firew,[4] Freweini Gebrehiwot,[4] Etenesh Gebreyohannes,[2] Zenebech Gella,[2] Addis Girma,[2] Zuriash Halefom,[2] Sorsa F Jama,[2] Annika Janson,[3,5] Binyam Kemal,[2] Abiy Kiflom,[1] Yidnekachew D Mazengiya,[2] Kalkidan Mekete,[4] Magdelawit Mengesha,[2] Meresha W Nega,[2] Israel A Otoro,[2] Joanna Schellenberg,[3] Tefera Taddele,[4] Gulilat Tefera,[1] Admasu Teketel,[1] Miraf Tesfaye,[2] Tsion Tsegaye,[1] Kidist Woldesenbet,[2] Yakob Wondarad,[2] Zemzem M Yusuf,[2] Kidist Zealiyas,[4] Mebratom H Zeweli,[2] Lars Åke Persson [ID],[3] Seblewengel Lemma [ID] [3]

For numbered affiliations see end of article.

**Correspondence to**
Dr Seblewengel Lemma;
Seblewengel.abreham@lshtm.ac.uk

## ABSTRACT

**Objective** A routine health information system (RHIS) enables decision making in the healthcare system. We aimed to analyse data quality at the district and regional level and explore factors and perceptions affecting the quality and use of routine data.

**Design** This was a mixed-methods study. We used the WHO toolkit for analysing data quality and interviewed staff at the point of data generation and along with the flow of data. Data were analysed using the Performance of Routine Information System Management framework.

**Setting** This study was performed in eight districts in four regions of Ethiopia. The study was nested within a 2-year programme of the Operational Research and Coaching for government Analysts.

**Participants** We visited 45 health posts, 1 district hospital, 16 health centres and 8 district offices for analysis of routine RHIS data and interviewed 117 staff members for the qualitative assessment.

**Outcome measures** We assessed availability of source documents, completeness, timeliness and accuracy of reporting of routine data, and explored data quality and use perceptions.

**Results** There was variable quality of both indicator and data element. Data on maternal health and immunisation were of higher quality than data on child nutrition. Issues ranged from simple organisational factors, such as availability of register books, to intricate technical issues, like complexity of indicators and choice of denominators based on population estimates. Respondents showed knowledge of the reporting procedures, but also demonstrated limited skills, lack of supportive supervision and reporting to please the next level. We saw limited examples of the use of data by the staff who were responsible for data reporting.

### Strengths and limitations of this study

► We assessed data quality and explored perceptions around data quality and use across a range of health indicators.
► Over 100 staff from different levels of Ethiopia's health system were interviewed and we attained thematic saturation.
► The qualitative findings suggested similar data quality problems as the quantitative results.
► We conducted a member check test, confirming that our results were credible.
► Our results from the quantitative data have limited generalisability, because we took a small sample size which was purposive rather than representative.

**Conclusion** We identified important organisational, technical, behavioural and process factors that need further attention to improve the quality and use of RHIS data in Ethiopia.

## BACKGROUND

High-quality, real-time data on the burden of disease and performance of the health sector are critical for decision-making and resource allocation.[1] A routine health information system (RHIS) aggregates information across the health system.[2–4] Despite improvements, efforts to increase coverage, quality, equity and accountability of health services are often hampered by the lack of reliable data.[5–7]

The Ethiopian Ministry of Health (MOH) named the Information Revolution as one

of four agendas in its first Health Sector Transformation Plan,[8] aiming to advance information collection, analysis, presentation and dissemination. RHIS data are generated at the point of service delivery at primary level (health posts, health centres, primary hospitals), secondary level (general hospitals) and tertiary-level healthcare (specialised hospitals). The web-based open-source computer software District Health Information System was introduced in 2015.[9 10] Data are forwarded and aggregated at district, regional and national administrative levels. However, the quality and use of RHIS data continues to be a challenge in Ethiopia[11–14] and elsewhere.[15–17]

Factors affecting data quality can be classified as technical, behavioural and organisational according to the Performance of Routine Information System Management (PRISM) framework. Technical factors relate to the ease of data collection, collation, analysis and reporting while behavioural factors include individuals' knowledge, attitude and skills related to RHIS processes. Organisational factors focus on availing human capital, infrastructure and a functional control system.[18] These factors directly affect RHIS performance but also interact with each other, requiring an integrated approach to produce favourable outcomes.[19] Understanding how these factors function at national level using the PRISM conceptual framework is an appropriate way to identify and implement appropriate interventions.

The overall aim of this study was to analyse RHIS data quality and use at district and regional levels, and explore perceptions of factors affecting data quality through a mixed-methods approach. This paper brings together findings from the operational research and coaching for analysts (ORCA) work at district and regional level to contribute to understanding and strengthening the RHIS across the whole health system. The specific objectives were to analyse the timeliness, completeness and accuracy of reporting of RHIS data generated at primary healthcare level, and to explore reasons for problems in data quality and use along the flow of data.

## METHODS
### Study setting and design
The Ethiopian MOH initiated the ORCA project in collaboration with the Ethiopian Public Health Institute (EPHI), the Ethiopian Pharmaceutical Supply Agency (EPSA) and London School of Hygiene & Tropical Medicine. ORCA was designed to guide participants through a research cycle that diagnosed and investigated the current state of data quality and use within the Ethiopian health system, taking into consideration key strategic health metrics. A group of 36 analysts from the MOH, EPHI and EPSA participated alongside their normal work duties from June 2018 to June 2020. The ORCA participants chose to work in six thematic groups: Maternal Health, Neonatal Survival, Immunisation, Child Nutrition, Malaria and Tuberculosis.

This was a mixed-methods study performed by the ORCA participants including quantitative analysis of district-level data, complemented by qualitative interviews with key informants at different levels. Fieldwork was conducted by each ORCA thematic group. Data were collected in eight districts in four regions in Ethiopia (Afar, Oromia, Southern Nations, Nationalities and People's region and Tigray), selected in consultation with the regional health offices, from August to December 2019.

### Sampling and recruitment
Health centres and health posts providing services for more than 1 year were included in the quantitative data collection. In each district, aggregated data were also assessed at district health offices. For the qualitative assessment, key informants were recruited purposively along the flow of data from health posts, health centres, district health offices, zonal health offices, regional health bureaus and the MOH. Informants had served for at least 1 year in their respective post, and could provide in-depth information about RHIS data. The informants' professional designations were health extension worker, head of health facility, RHIS focal person, head of district health office and programme expert at district, zonal, regional or federal level.

### Data collection and processing
Each ORCA thematic group prepared a desk review checklist for relevant indicators, drawing on standard data quality assessment tools[20] (online supplemental files 1–4). The checklists were pretested in similar settings. Data were collected at health facilities from primary source documents and entered into Microsoft Excel for analysis.

A qualitative topic guide was prepared in English by each thematic group and translated into local languages (Amharic, Oromiffaa, Tigrigna or Afar). Interview guides were pretested and refined, and further adapted during fieldwork to improve comprehensibility. Data collectors were ORCA team members trained in qualitative and mixed-methods research. Interviews lasted 30–60 min, recorded, and field notes were taken by group members. After data collection, group members reflected on their work and identified points for exploration during subsequent interviews. Recordings were transcribed verbatim. Ten per cent of the transcripts were cross-checked with the audio for completeness and accuracy.

### Quantitative information
All definitions were based on the WHO toolkit.[20] Details of the toolkit were discussed in our previous similar work[11]; Availability of source document and report was presented as a percentage, that is, facilities with records, divided by the total facility months investigated; Completeness of reporting indicated the percentage of monthly reports received by the next level; Timeliness of reporting covered the proportion sent on time; and Accuracy of reporting

indicated the ratio of numbers recounted and classified as exact match, within the data quality range (0.9–1.1), over-reporting (<0.9) or under-reporting (>1.1). Results were categorised by type of indicators and presented as percentage of health facility months.

## Qualitative data analysis

Each group conducted thematic content analysis. After reading the verbatim transcripts, all group members coded the same interview and agreed on a coding framework. The group members divided interviews among themselves for coding, and met regularly to add codes to capture emerging ideas. Groups categorised codes into broader thematic areas. Each group prepared a report on qualitative results that were shared across groups. The joint results from all six thematic groups were synthesised using the PRISM framework[18] (online supplemental file 5). Regular discussions were held to reflect on similarities and differences across the data sets, check for outliers and contradictory findings, and agree on distribution of key themes within the simplified structure of the framework. Finally, the result was shared with seven respondents at MOH to check for credibility.

## Patient and public involvement

Patients or the public were not involved in the design, or conduct, or reporting, or dissemination plans of our research.

## RESULTS

In total, 62 facilities and eight district health offices were visited for analysis of RHIS data and 117 key informants were interviewed (table 1). Of all interviews, 35/117 (30%) were with health extension workers at health posts.

## Availability of source documents, completeness, timeliness and accuracy of reporting

The availability of source document ranged from 55% to 100%. Only documents for skilled birth attendance reached 100% in observed health facilities (figure 1).

The majority of indicators had gaps in reporting. Maternal health and postnatal indicators had the most gaps in reporting. Completeness of reporting for nutrition was also low, at slightly over 50% for the facility-months reviewed. Completeness was much higher for immunisation. Timeliness was over 90% for maternal health indicators, whereas just over half of reports for nutrition indicators were submitted on time (figure 2).

Maternal and immunisation indicators had lower proportions of reports within the range for acceptable quality, whereas nutrition indicators were mainly reported within the quality range. Varying levels of over-reporting were observed in all service coverage indicators, but not for severe acute malnutrition (figures 2 and 3).

## Respondents' views on data processes and quality

Interview respondents reported that data generation and flow mostly occurred as intended. At health facilities, data

**Table 1** Desk reviews and qualitative interviews conducted by ORCA thematic groups and other background information, Ethiopia, 2019/2020

| Characteristics | Desk review (n=70) | Qualitative interviews (n=117) |
|---|---|---|
| Gender | | |
| Male | | 75 (64%) |
| Female | | 42 (36%) |
| Health facilities/offices visited | | |
| Health post | 45 (64%) | 35 (30%) |
| Health centre | 16 (23%) | 33 (28%) |
| District hospital | 1 (1%) | 2 (2%) |
| District health offices | 8 (11%) | 21 (18%) |
| Zonal health office | 0 (0%) | 1 (1%) |
| Regional health office | 0 (0%) | 17 (15%) |
| Federal ministry of health | 0 (0%) | 8 (7%) |
| Region | | |
| Tigray | 5 (7%) | 15 (13%) |
| Afar | 19 (27%) | 37 (32%) |
| Oromia | 17 (24%) | 25 (21%) |
| SNNPR | 29 (41%) | 32 (27%) |
| National | 0 (0%) | 8 (7%) |
| Thematic group | | |
| Maternal health* | 12 (17%) | 18 (15%) |
| Neonatal Survival† | 17 (24%) | 14 (12%) |
| Immunization‡ | 9 (13%) | 12 (10%) |
| Child nutrition§ | 9 (13%) | 25 (21%) |
| Malaria¶ | 6 (9%) | 17 (15%) |
| Tuberculosis** | 17 (24%) | 31 (27%) |

*First antenatal care, fourth antenatal care, postnatal care and skilled delivery.
†Early institutional death (0–6 days), early community death (0–6 days), live birth in Kebele.
‡Pentavalent vaccine third dose, Measles, fully vaccinated.
§Vitamin A supplementation, deworming, severe acute malnutrition, growth monitoring promotion.
¶Suspected malaria, positive malaria, all malaria.
**New and relapse tuberculosis, and treated tuberculosis.
ORCA, operational research and coaching for analysts; SNNPR, Southern Nations Nationalities and Peoples Region.

were usually recorded by hand using standard on paper forms, while district health offices were more likely to use computers.

> There is already an established database up to Ministry of Health. Here in the District, it is totally electronic and we do not send data to the next level with a hard copy. Hard copy is only sent from lower level up to District level. (Focal person)

Data were compiled mainly for reporting to the next level, with the exception of health centres, where

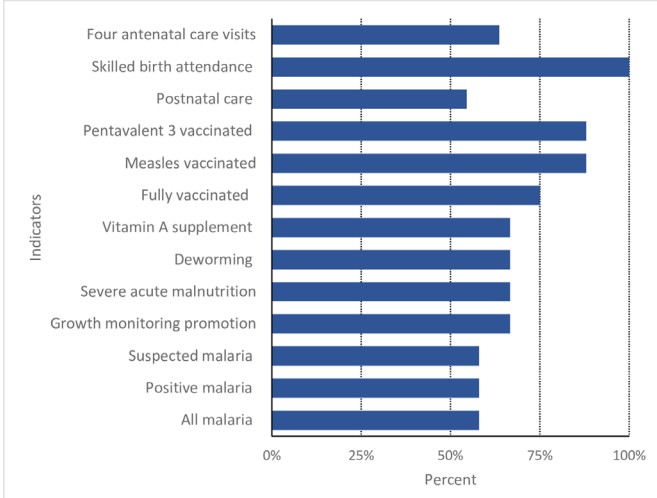

**Figure 1** Availability of source documents and reports for the facility-months observed.

performance monitoring teams used data to monitor health service delivery. Little was done to triangulate different sources of data in the system. For instance, logistics data on drug consumption were merely used to validate the service delivery report.

EPSA only knows consumption data and doesn't have patient data. It only compares what is supplied and what is consumed. Therefore, it is difficult to compare the discrepancy (Administrative staff)

### Data quality check
Respondents described a formal approach to data quality checks, that is, standard tools and procedures used to check RHIS data. This process addresses data quality attributes such as reporting timeliness and accuracy.

Recorded data, report, register, and tally are cross-checked. If the three are equal, we said the data are quality… Based on this the quality of data will be

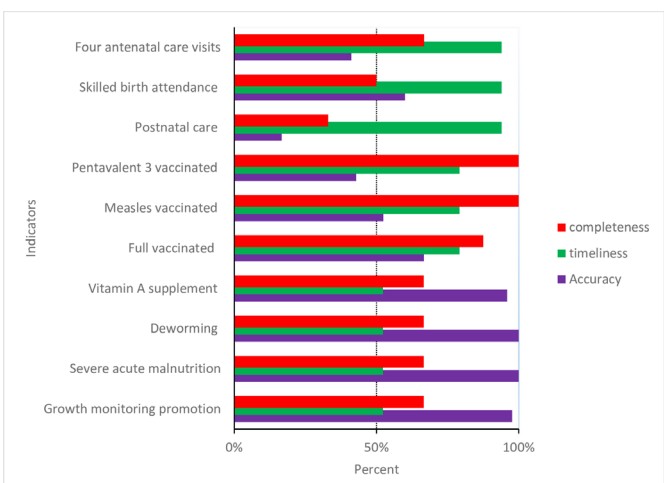

**Figure 2** Completeness, timeliness and accuracy of reporting for selected indicators in the routine health information system.

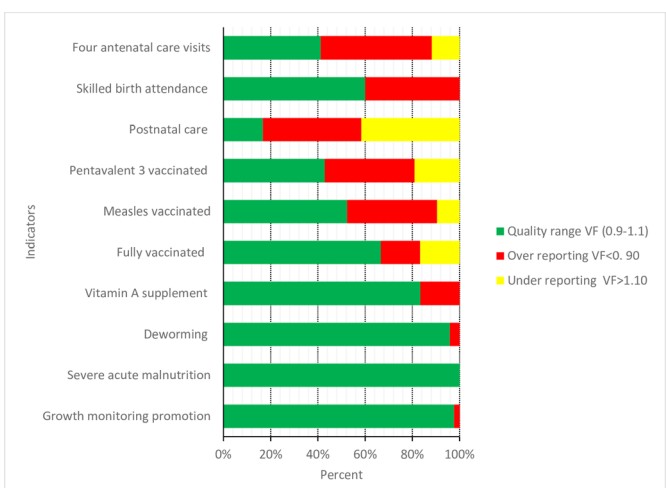

**Figure 3** Accuracy of reporting for selected indicators in the routine health information system. VF, Verification Factor.

ranked … The report and register will be checked for the specified period for each month. (Administrative staff)

Data and reports were verified before being sent to the next level either through phone call or in person review. This approach was reported to be more common than use of standard tools for checking data quality. Respondents said challenges come from lack of transportation, or competing demands on time.

As soon as the report is finalised, the health centre immediately reports to the district without any verification by the performance management team and the district health office then immediately send it to zone health office without a review. This is due to other competing priorities. (Focal person)

Sometimes reports were amended without consulting the source:

We will call and ask them to clarify. Most of the time, their phone will not work. Now for instance if they reported PCV 1 as zero or left it blank, I will take the figure of penta 1 because it is the same. I will take all antigens reported as first dose and third dose and fill the missing part. (Focal person)

### Perceived quality of RHIS data
Most respondents agreed that the RHIS data lacked consistency and were reported late. Lack of consistency was attributed to incorrect recording, modification or manipulation of data to compensate for the lack of data or resulting from poor understanding of the RHIS process (table 2).

Respondents mentioned several reasons for inadequate data quality, presented below as technical, organisational or behavioural factors.

**Table 2** Perceived data quality as reported by the respondents, Ethiopia, 2019/2020

| Perceived data quality issue | Illustrative quote |
| --- | --- |
| Data not recorded on time | 'Staff fill the registration over night when they have information that the supervisors from district health office will come.' (Focal person) |
| Wrong recording | 'The patients are taking drugs but are not reported. This creates under reporting. On the other hand, sometimes there is a practice of reporting patients of other diseases' (Health care provider) |
| | 'The health extension workers may include and report to us information which is not found in their tally sheet or register. That is what we evaluated.' (Focal person) |
| Double counting | 'Yes! There is double reporting in ANC. They are confusing. I mean …ehh…if they did not understand well each other, who didn't go, who comes there (health centre & Hospital), who is referred, they might report twice. A mother who just got a first ANC service there (health post) and comes for second service (health centre/Hospital) is also reported as first ANC again…' (Administrative staff) |
| Data manipulation | 'For instance, nutrition indicators are mostly reported as zero from the health post but sometimes we (the health centre) just put numbers that we think is appropriate by evaluating the health posts previous performance. And sometimes we get reports that are left blank and we just assume that as being zero and we fill the space with '0'.' (Data manager) |
| | 'So far, we did not come across (any) neonatal death report. However, I could not say there is no neonatal death at all…The weakness here is the death is not correctly reported.' (Data Manager) |
| Delayed reporting | 'The report doesn't come on time, for example the report is closed on 20th and from health posts it will be sent to us from 20th to 22th, we, in turn, we aggregate the health posts report including our health facility and we send the report to district until 26th of every month'. (Data manager) |
| | 'I would say the data has quality although there is a gap in timeliness. For example, one health post in our catchment area is relatively difficult for transportation. Due to that their report gets delayed for three or four days.' (Focal person) |

## Technical factors

Respondents expressed concern about the number and complexity of forms. Parallel reporting posed additional burdens on the system and contributed to poor data quality, occurring because some indicators that are relevant to several programmes were not captured centrally in the RHIS.

> Many partners need reports from us. Their data needs are different… The parallel report is still a problem and ignorance is there, in the higher level (Administrative staff).

Understanding indicators varied between respondents. Maternal health indicators such as first and fourth antenatal care visits were considered challenging, with additional complexity due to wrongly including information on gestational age:

> ANC1 is a visit by a woman for the first time. A pregnant woman within 16-24 week of gestational age is ANC1. (Health care provider)

> … starting from the first visit, if a pregnant mother comes 28 week for the second, and 32 week for third, eeh … 36 for third time consecutively and comes again from 36 to 40th week, I take her last visit as ANC4. (Health care provider)

Understanding RHIS indicators was also limited by language issues as not all forms and job aids were translated into local languages. This posed a challenge especially at the health post.

> The problem is [the] integrated card and even [the family] folder is difficult to understand since it is in English (Health care provider)

> The Amharic version [of EPI card] was printed and distributed. How could the people do the work? Those down there [at health posts] do not understand Amharic. (Administrative staff).

Another cross-cutting technical issue was inappropriate denominators used for calculating health service coverage. Targets for different services were based on population estimates using the last available census from 2007. Thus, set targets can differ from actual numbers of individuals requiring the service (eg, pregnant women or children eligible for vaccines) in a district or catchment.

> We are mostly being challenged by this [denominator issue]. For example, there is one kebele which was given a target of 46 for ANC service based on the population conversion factor, but there are only 18 pregnant mothers found in the kebele. (Administrative staff)

Not all health facilities had access to computers, but where health centres had computers and internet access and in most districts, reports were sent online. This was considered progress despite significant variations in use of technology.

Out of the three health centres, one of them submits its report online. It is 22 km away from here, they have electricity but there was no connection, now the zonal health department provided them 3G CDMA [Code Division Multiple Access) and they are using that. The other two submits offline using a flash disk. (Focal person)

## Organisational factors

The RHIS uses nationally developed standard forms and registers. Selected service registration books come from the ministry, while remaining forms are sent from regional, zonal and district health offices. Shortage of supplies such as registration books, tally sheet and other forms were repeatedly mentioned.

For example, now there is no tally sheet for postnatal, and even a registration book…it is not available in the district either. We are using attaching papers as register; we can show you … (Health care provider)

The district office diverted resources allocated to other activities or duplicated forms to address supply gaps. It was not uncommon for healthcare workers to use their own money.

…. budget is not allocated separately for activities related to health information, this is a problem in our district and it is also a problem in our zone, there is no direct budget allocated for this, we use from other funds that we get from aid. (Administrative staff)

Limited electricity, computers and transportation often affected health posts. At health centres, frequent interruption of power coupled with lack of backup affected timely reporting, and availability of forms for registration and reporting. Table 3 lists resource and infrastructure challenges reported by respondents.

Except in a few health facilities, health workers were responsible for RHIS activities in addition to their clinical work. Human resource shortages were more prominent at health post level where one or two health extension workers provide more than 16 health service packages and produce reports for each. This workload was said to contribute to poor data quality.

… because what comes from the districts puts pressure on us [health extension workers]. What comes from the (kebele) cabinet brings pressure on us [health extension workers]. There are times we even do agricultural activities, which doesn't concern us so it is very difficult. And when it is time to work on report, there are a lot of forms to fill and it is difficult for us. (Health care provider)

There was a clear demand for training although a few respondents mentioned that training hadn't posed problems. Where training was lacking, staff turnover was mentioned as the main cause. Moreover, recent changes to RHIS tools called for more training.

Even we have no a clear understanding on the data element in the DHIS-2, the data elements are so many, it is not user-friendly. There is confusion among us which data element to use and the District level supervisor seems clueless on this issue as we have witnessed during the recent supervision (Focal person).

District health offices supervise and support health centres in the district, and each health centre does the same for health posts in its catchment area. There is also a performance monitoring team at the health centre that

**Table 3** Resource and infrastructure related challenges as reported by respondents, in Ethiopia, 2019/2020

| Resource constraint | Illustrative quote |
|---|---|
| Lack of transportation | *'Transportation is our biggest challenge. In the summer season, sometimes we can't send the report. It is difficult to cross the rivers. We try to cross by walking. Once when I was crossing the river, I lost my report papers by the flood.'* (Health care provider) |
| Lack or interruption of electricity | *'Especially [when] a report gets delayed; there is no backup, this power is not how you see it, sometimes when it interrupts it's not fixed soon; because of this, when power is off, everything disrupts, even we can't print; we can't send the report.'* (Administrative staff) |
| No computer | *'It was not possible to send report using CDs (compact discs) as there were no computers in some places.'* (Focal person) |
| | *'There are a lot of Health centres that have no computer, and even those who have computers, some of them have no electricity.'* (Focal person). |
| Printer | *'Having printer is a problem, we [HEWs] can't get printed reporting forms when we need them, and it is not always available.'* (Health care provider) |
| Poor access to internet | *'…Even in the areas where the online system is launched there is an internet problem. So generally, theoretically we are shifted to digitalization, [but] practically there is no enabling condition to digitalization.'* (Focal person) |
| | *'Since there is no regular telecommunication cable line we use offline; unfortunately, we have taken the computer to the district for installing the offline application and…, we believe its electronic based on the District health information system 2(DHIS2).'* (Administrative staff) |

should provide regular feedback to health centres and health posts. However, supervision was said to be infrequent and not always supportive.

> They came once or twice per year. In the last three months, no one came to our health post from health centre or district [district] or zone. (Health care provider)

Supervision was said to rarely focus on data quality. Furthermore, supervisory staff were considered inexperienced in providing technical support on data quality to lower level staff.

> The support focuses on technical coaching on the [health] service, but not on the data quality (Health care provider)

The local performance monitoring team serves as a check-and balance system; it monitors the service delivery output and provides the necessary support to improve performance as well as data quality. However, several respondents reported that the team met infrequently and was suboptimal.

Respondents, including administrative staff, believed there ought to ways to holding people accountable when data quality is compromised. It was felt over-reporting of health service coverage rewards health facilities, which are seen to achieve targets without anyone confirmation of reporting accuracy.

> If there is any reward planned from the higher level, it will go directly to those who reported higher coverage. When additional budget is assigned, the district with higher coverage is given priority. Other districts see this and inflate their coverage to get the same advantage and never report the actual figures. (Focal person)

There was also fear of reporting low service coverage or unwanted results such as neonatal death, leading to data manipulation to please higher-level administrative staff.

> I want to report the actual figures, by the way I am happy when you told me to interview me without the presence of my boss, because it is hard to explain in his presence. For instance, there is an intention to over report delivery service and decrease or report zero for still births and the like. (Focal person)

### Behavioural factors
Gaps in knowledge and skills related to the RHIS process was expressed by administrative staff and some healthcare providers, including difficulties understanding the registration and other forms, performance management, and basics of data entry and analysis. In addition, lack of knowledge and skill on checking data quality was reported.

> We do not have information and skill on how to work on the quality of data and we have limited knowledge

on how to work on performance management, comparison and so on. (Focal person).

Health workers repeatedly mentioned lack of interest in RHIS resulting from low personal motivation and work overload.

> Sometimes we get fed up, because the format asks for too many things and we don't understand, we say: -What? We don't fill it and we submit without filling the information (Health care provider).

### Perceived use of data
A culture of data use was not well developed and the utility of generating data routinely not well understood.

> The purpose of the analysed health data is for decision making, this is the fact, but still there is a gap in using the data. It should be good if the stakeholders of the health facility use the analysed data. (Administrative staff)

> Drugs are distributed to health posts monthly depending on the consumption status and we (Logistic focal) don't give them unless they report number of cases. Otherwise drugs will expire there…(Focal person)

Data use for programming was appreciated more at higher levels of the health system. It was reported that data were used for monitoring performance and identifying gaps during annual planning or to manage drug supply. There were also initiatives as reported by administrative staff to improve data use.

> I believe that conducting data verification regularly at lower level and provide close support to the Health centre and Health post staff will help to improve data quality and use problem (Focal person).

> We (nutrition expert) use HMIS data; we found over reporting and lack of reporting sometimes, conducted performance reviews; our data source [was] HMIS, besides, we use the nutrition data base as alternative source of information…Data utilization [is] better at woreda health office where nutrition experts are available. (Focal person)

### DISCUSSION
We assessed quality of RHIS data in Ethiopia across multiple health indicators and explored reasons affecting quality, from data generation through to reporting and use. We observed variations in quality between indicators. Whereas there was timely reporting of some indicators but with less accuracy, others were reported accurately, but not on time or completely, adding to concerns about RHIS data quality and utility. Determinants of data quality ranged from simple logistical issues, such as supply of registry books, to complex technical issues, such as the size of a target population

used as the denominator to calculate coverage. Organisational factors related to training and supervision stretched into more complex behavioural issues of motivation and fear of reporting unfavourable events.

One strength of this study is that we interviewed over 100 informants representing a mix of staff in the health system and achieved thematic saturation, suggesting our findings have relevance throughout the Ethiopian health system. We also tested the credibility of our result using a member check approach and confirmed the results. A potential limitation of this study was the small quantitative assessment sample; however, this part of the study was designed to prepare the background for the in-depth qualitative assessment rather than to yield statistically representative results. Our qualitative findings reflected similar data quality problems. The data quality assessment tool we used may not rule out mistakes or wrong reporting in the RHIS processes. It could be argued that PRISM framework used to guide our analysis may not clearly delineate some of the factors to either behavioural or organisational factor.

Both quantitative and qualitative results confirmed limited availability of source documents. Availability varied by indicator, and only one indicator had source documents for the whole observation period. Respondents described registration book and tally sheet shortages.

Completeness, timeliness and accuracy of reporting were found to be inadequate for selected key indicators. Endriyas et al showed a similar pattern of variability of accuracy among indictors in Ethiopia,[7] with maternal indicators exhibiting better quality. This may result from a national focus on maternal and child health services. Endriyas *et al* and other studies have also described over-reporting of service coverage and under-reporting of disease similar to our findings.[21–23]

Complexity of registration forms and language barriers detrimentally affect accurate data recording.[23 24] While inadequate knowledge of RHIS is a cross-cutting issue, it proved more problematic at lower levels of the health system, where data are generated. Other studies report that not understanding indicators[25] and poor competency in recording[26] affect data quality.

Human resource shortages appeared to affect all levels of the RHIS process, most prominently at health facilities, where health workers are responsible for data collection on top of their clinical service. This creates workload and reduces motivation for RHIS. Similar human resource challenges have been found elsewhere.[7 12 23 26] Furthermore, access to technology that might ease this workload remains low. Disruption and shortages of data collection forms and registration books also contributed to delayed or inaccurate recording. Others have found that simplified data collection forms or digital tools can reduce the RHIS burden[27] and improve data quality.[28 29]

The delay in data transmission emerged as a common problem at health facility level. As mentioned above, access to technology such as computers and internet would improve timely data transmission, although this would not address the problem of parallel reporting requirements that also add to workload and reporting delays, as cited by Gebreslassie et al.[30]

Data processing and analysis occurred primarily at higher levels. Gaps in knowledge and skill were reported to challenge these processes in other settings.[26 31] Use of outdated population data for denominators has already been raised as a concern in previous analyses of Ethiopian RHIS.[11] Similarly, inconsistency of denominators used to estimate coverage was reported by Bosch-Capblanch *et al.*[21]

Although data quality checking and feedback systems using standard tools exist, these are rarely implemented. Other studies have noted this determinant of poor data quality,[7 23 30 32] and have shown that regular data quality assurance with appropriate feedback can motivate positive changes in data quality and use.[16 33] What was unique in this study was the establishment of performance monitoring teams to oversee activities in the health system including data quality, but lack of budget and gap in skills negatively affected the functionality of this mechanism.

Although staff fear reporting unfavourable data, we nonetheless found hopes for a system that holds health workers and health facilities accountable for generating inaccurate data, even while long-term challenges persist. Respondents may have recognised elements of 'blame culture' in the Ethiopian RHIS, described by others as emerging where hierarchical management structures reward compliance over efforts to expose poor quality and function.[34] The result is that staff eschew negative attention, which does not predispose them to raise awareness of systemic weaknesses or help develop genuine accountability.

In terms of data use, this was uncommon at sites of data generation although administrative staff did employ local data for planning and monitoring local performance. Similar findings were reported elsewhere.[14 22] Many studies have recognised the effect data use and data quality have on one another.[4 7 16 35]

In summary, many factors negatively affecting data quality persist within Ethiopia's RHIS. Some of these factors could be tackled with existing resources, such as ensuring availability of registration forms and tally sheets in local languages. On-the-job training for healthcare workers at the lower level can boost their knowledge and skills, but also their motivation. Strengthening the existing data quality and feedback system is essential. Human resources for RHIS, infrastructure and budget are cross-cutting factors that affect the whole RHIS process and require longer-term planning and multisectoral engagement, as does introducing a work culture that values proactive challenges to existing weaknesses. More qualitative work on data use could help understand barriers that could be tackled.

## Consent

Permission was obtained to conduct the fieldwork from each regional health office, district health office and health facility visited prior to data collection. Written informed consent was obtained from all participants and measures taken to ensure anonymity. Translators were not chosen among supervisory staff or others on whom the respondent could be dependent. Staff categories were expressed in general terms, such as 'administrative' to ensure anonymity.

**Author affiliations**
¹Ethiopian Pharmaceutical Supply Agency, Addis Ababa, Ethiopia
²Ethiopia Ministry of Health, Addis Ababa, Ethiopia
³London School of Hygiene & Tropical Medicine, London, UK
⁴Ethiopian Public Health Institute, Addis Ababa, Ethiopia
⁵Department of Women's and Children's Health, Karolinska Institutet, Stockholm, Sweden

**Acknowledgements** The authors thank the study participants for taking part in this research process and shared their experience. The authors are grateful for the logistic support received from LSHTM Ethiopia team, and Ethiopian public health institute.

**Contributors** All members of the ORCA team participated in designing and conducting the study. The following authors drafted the manuscript collaboratively: AA, TMA, MMA, HAA, ESA, DB, NB, MG, TB, JB, EC, MD, THD, AD, HSF, FG, EG, ZG, AG, ZH, SFJ, AJ, BK, AK, YDM, KM, MM, MWN, IAO, JS, TeT, GT, AT, MT, TsT, KW, YW, ZMY, KZ, MHZ, LAP and SL. All authors contributed in the later steps of the writing process and approved of the final manuscript and agreed to be accountable for all aspects of the work. SL is responsible for the overall content as a guarantor.

**Funding** The ORCA project was funded by the Bill and Melinda Gates Foundation with a grant to the London School of Hygiene & Tropical Medicine, grant number: INV-010320.

**Competing interests** None declared.

**Patient consent for publication** Not applicable.

**Ethics approval** This study involves human participants and the ORCA thematic groups' proposals were reviewed and approved by the Ethiopian Public Health Institute, Institutional Review Board Reference number: EPHI-IRB-188-2019, EPHI-IRB-196-2019, EPHI-IRB-190-2019, EPHI-IRB-202-2019, EPHI-IRB-189-2019, EPHI-IRB-2014-2019.

**Provenance and peer review** Not commissioned; externally peer reviewed.

**Data availability statement** Data are available on reasonable request.

**ORCID iDs**
Habtamu A Anteneh http://orcid.org/0000-0003-3755-3357
Emiamrew S Ayalew http://orcid.org/0000-0002-4857-5802
Della Berhanu http://orcid.org/0000-0002-4984-893X
Lars Åke Persson http://orcid.org/0000-0003-0710-7954
Seblewengel Lemma http://orcid.org/0000-0001-5910-3723

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
