## [Reviewer comments · BMJ Open]

ARTICLE DETAILS

TITLE (PROVISIONAL)	Exploring data quality and use of the routine health information system in Ethiopia: a mixed-methods study
AUTHORS	Adane, Abyot; Adege, Tewabe M.; Ahmed, Mesoud M.; Anteneh, Habtamu A.; Ayalew, Emiamrew S.; Berhanu, Della; Berhanu, Netsanet; Getnet, Misrak; Bishaw, Tesfahun; Busza, Joanna; Cherinet, Eshetu; Dereje, Mamo; Desta, Tsega H; Dibabe, Abera; Firew, Heven S.; Gebrehiwot, Freweini; Gebreyohannes, Etenesh; Gella, Zenebech; Girma, Addis; Halefom, Zuriash; Jama, Sorsa F.; Janson, Annika; Kemal, Binyam; Kiflom, Abiy; Mazengiya, Yidnekachew D.; Mekete, Kalkidan; Mengesha, Magdelawit; Nega, Meresha W.; Otoro, Israel A.; Schellenberg, Joanna; Taddele, Tefera; Tefera, Gulilat; Teketel, Admasu; Tesfaye, Miraf; Tsegaye, Tsion; Woldesenbet, Kidist; Wondarad, Yakob; Yusuf, Zemzem M.; Zealiyas, Kidist; Zeweli, Mebratom H.; Persson, Lars; Lemma, Seblewengel

VERSION 1 – REVIEW

REVIEWER	Valls Martínez, María del Carmen Universidad de Almería, Economics and Business
REVIEW RETURNED	10-Apr-2021

GENERAL COMMENTS	Review of the article 2021-050356 “Exploring data quality and use of the routine health information system in Ethiopia: a mixed-methods study” The article analyses the data quality of the health system in Ethiopia, through direct interviews. The aim is really interesting but the work lacks of several flaws, in my opinion. First, the authors should have added the survey as an appendix to the paper, so that the reader can better understand the work done. Second, the description of the entire process is too extensive, taking up most of the article. It could have been explained briefly in order to focus the work not on the interview process but on the results achieved. Third, in my view, the survey should have been conducted on the basis of a series of Likert-scale questions. In this way, much more in-depth analytical results could have been extracted and statistical techniques could have been applied to support the results that I consider to be poor. For example, there could have been remarkable similarities or differences between the different thematic groups: maternal health, neonatal survival, immunization, child nutrition, malaria and tuberculosis. Unfortunately, I consider that the article presented is not of sufficient quality to be published in this journal.
---

REVIEWER	Pisani, Elizabeth King's College London
-----------------	--

GENERAL COMMENTS

Thanks for the opportunity to review the paper "Exploring data quality and use of the routine health information system in Ethiopia: a mixed-methods study". The paper describes the results of an investigation of the quality of routine health information reported in 8 districts in four regions of Ethiopia. It then focuses largely on describing the obstacles to better system functioning, based on information reported in qualitative interviews.

Framing and import

The paper is somewhat normative in its framing. Its opening sentence declares that "High-quality, real-time data on the burden of disease and performance of the health sector are critical for decision-making and resource allocation". However, as the introduction goes on to say, such data are actually often absent -- the paper goes on to make clear that this is also the case in Ethiopia.

The fact is, decisions get made and resources get allocated regardless of whether the data exist or not; indeed, again as described in the findings, the data are very often NOT used for decision-making or resource allocation, even where they do exist. The discussion makes mention of this (in the paragraph beginning p27 line42), referring almost in passing to the critical interaction between local use and data quality.

The study benefits from a relatively large number of interviews with people at various levels of the health system. The minimal information presented on data use suggest that:

- a) at the local level, mis-reporting is common and (perhaps because reported data are known to be unreliable?) reported data are rarely used in planning
- b) data reported up from the local level are more commonly used at higher levels -- sometimes after being 'completed' based on unsubstantiated assumptions (perhaps because people at those levels are less aware about how unreliable the data are?).

It would be of great interest if the authors could dedicate more time to exploring in greater depth what respondents have to say about the actual or perceived utility of data; then in the discussion reflect on what this means for health policy in Ethiopia.

Methods

The section describing quantitative methods relies on the readers' familiarity with the WHO toolkit, which cannot be taken for granted. More detail would be welcome. As presented in the methods, three of the 4 (completeness, timeliness and accuracy) measure inter-level reporting processes, rather than quality per se: i.e. they measure what goes in and comes out, but not the extent to which the quality of what goes in (i.e. the records kept at facility level) accurately represents what happens at the facility. This is an occupational hazard for those trying to do retrospective data quality assessments, but given the prominence in the qualitative data of evidence of active mis-reporting, it at the very least merits some discussion. It would be especially interesting to know how evidence of mis-reporting is collected in sties that use standardised electronic data forms, since these are designed automatically to aggregate the data entered for reporting to the next level. Also, out of curiosity, how do you deal methodologically with the retrospective data completion described on p16 line6?

Does the higher level get scored for over-reporting?

Re qualitative data coding: the methods describe parallel coding and synthesis of a framework (p12 line29), which was then iteratively developed. What was the original framework based on?

	Since evaluations of HIS are not uncommon, it would be helpful to future researchers if the authors could share their final coding framework as supplementary material. Results Please provide the interviews in Table 1 by gender (and, for the fun of it, consider whether gender representation in service delivery may have some bearing on the variance of "quality" of reporting of different indicators reported mentioned on p16 line3) Re structure of the analysis: I understand that the authors are conducting their thematic analysis according to the PRISM framework. One limitation of that framework is that it does not sufficiently consider the role of incentive structures, both institutional and personal. Analysts thus struggle to classify important systemic drivers of poor performance (such as: "I'm already working a 12 hour day in a stressful situation to save lives for low pay, and now you want me to fill the same information into three spreadsheets to that you can report to different donors and maybe, if we've had a bad month or you've overcalculated the denominator, dock our budget for not meeting targets? No chance!") into 'organisational' or 'behavioural'. The paper rightly underlines the interplay between the PRISM categories (p9 line39); incentive structures are very often the thread that links them, and merit more explicit consideration, at least in the discussion. A consideration of what the 117 interviews through the lens of incentives might also shed some light on the reasons why some of the most obvious and frequently-cited recommendations about health information systems have not actually been implemented in Ethiopia, despite the nominal importance of the Information Revolution in the government's health plan. Many of these are mentioned in the paper; they include such obvious and well-rehearsed elements as availability of reporting forms in appropriate languages, reduction of burden of reporting at the facility level, prioritisation of data for which there is local utility and demand, and ring-fenced funding for data systems.
--	--

VERSION 1 – AUTHOR RESPONSE

Reviewer: 1

Dr. María del Carmen Valls Martínez, Universidad de Almería

Comments to the Author:

Review of the article 2021-050356 "Exploring data quality and use of the routine health information system in Ethiopia: a mixed-methods study"

The article analyses the data quality of the health system in Ethiopia, through direct interviews. The aim is really interesting but the work lacks of several flaws, in my opinion.

Reviewer 1.1: First, the authors should have added the survey as an appendix to the paper, so that the reader can better understand the work done.

Authors' response:

Thank you for your insight. Yes, rightly said we did both quantitative and qualitative data collection. The quantitative data was mainly document review of the routine health information system and we used checklists to collect data. We now included the quantitative desk review checklists as a supplementary material.

Reviewer 1.2: Second, the description of the entire process is too extensive, taking up most of the article. It could have been explained briefly in order to focus the work not on the interview process but on the results achieved.

Authors' response:

One of the journal's requirements is for authors to provide adequate information on the research process based on a predefined theme. Our capacity-building project was a two-year journey where facilitating the process was part of the research experience, with relevance to other settings. We tried to be as transparent as possible regarding the full methodology we followed, as we feel this is a finding in itself.

Reviewer 1.3: Third, in my view, the survey should have been conducted on the basis of a series of Likert-scale questions. In this way, much more in-depth analytical results could have been extracted and statistical techniques could have been applied to support the results that I consider to be poor. For example, there could have been remarkable similarities or differences between the different thematic groups: maternal health, neonatal survival, immunization, child nutrition, malaria and tuberculosis.

Authors' response:

The main objective of the quantitative desk review was to objectively measure the data quality for the selected indicators. For the quantitative data, the team reviewed available routine data based on a standardised data quality assessment checklist. To use Likert scales to answer the objectives each team set would have been an alternative approach to the open-ended questions in the qualitative parts, and we agree that Likert scales could have enabled comparisons. However, we used interviewing with open-ended questions which can help explore in-depth. The ORCA project aimed for a deeper understanding of reporting practises and we wanted to encourage a reflective attitude on the actual behaviour to help explore the phenomena in-depth.

Reviewer: 2

Dr. Elizabeth Pisani, King's College London

Comments to the Author:

Thanks for the opportunity to review the paper "Exploring data quality and use of the routine health information system in Ethiopia: a mixed-methods study". The paper describes the results of an investigation of the quality of routine health information reported in 8 districts in four regions of Ethiopia. It then focuses largely on describing the obstacles to better system functioning, based on information reported in qualitative interviews.

Reviewer 2.1: Framing and import

The paper is somewhat normative in its framing. Its opening sentence declares that "High-quality, real-time data on the burden of disease and performance of the health sector are critical for decision-making and resource allocation". However, as the introduction goes on to say, such data are actually often absent -- the paper goes on to make clear that this is also the case in Ethiopia.

Authors' response:

Thank you for your encouraging words and for acknowledging the aim of our study. We agree that there is a somewhat normative frame to our work. We were not neutral to the need of high-quality data on health system performance.

Reviewer 2.2: The fact is, decisions get made and resources get allocated regardless of whether the data exist or not; indeed, again as described in the findings, the data are very often NOT used for decision-making or resource allocation, even where they do exist. The discussion makes mention of this (in the paragraph beginning p27 line42), referring almost in passing to the critical interaction between local use and data quality.

The study benefits from a relatively large number of interviews with people at various levels of the health system.

The minimal information presented on data use suggest that:

a) at the local level, mis-reporting is common and (perhaps because reported data are known to be unreliable?) reported data are rarely used in planning

b) data reported up from the local level are more commonly used at higher levels -- sometimes after being 'completed' based on unsubstantiated assumptions (perhaps because people at those levels are less aware about how unreliable the data are?).

It would be of great interest if the authors could dedicate more time to exploring in greater depth what respondents have to say about the actual or perceived utility of data; then in the discussion reflect on what this means for health policy in Ethiopia.

Authors' response:

Thank you for reviewing it in such a depth. We agree that data is under-utilized, likely for a number of reasons, many of which you suggest. We also agree on the interlinkage of data quality and data use. In the monitoring and evaluation standard practise, much attention is given to accuracy along the line of reporting whereas the ORCA project aimed to use methods to explore further. Although data use was given equal emphasis from the very beginning of the study, this issue didn't come-out well in the interviews. We do not know the reasons for this, but could speculate that issues of local ownership and data use were not really in the mindset of the persons interviewed.

During interviews, respondents emphasized issues related to the quality of data and their concerns around it. This surely requires further investigation in future similar works, and we now included it in our recommendation as one of the area that should be explored as future qualitative research (page 27, line 19,20). As:

"More qualitative work on data use could help understand barriers that could be tackled."

We have added a few quotes on (page-23,24) that detail about data use:

"Drugs are distributed to health posts monthly depending on the consumption status and we (Logistic focal) don't give them unless they report number of cases. Otherwise drugs will expire there..." (Focal person)

"We (nutrition expert) use HMIS data; we found over reporting and lack of reporting sometimes, conducted performance reviews; our data source [was] HMIS, besides, we use the nutrition data base as alternative source of information. Data utilization [is] better at woreda health office where nutrition experts are available." (Focal person)

Reviewer 2.3: Methods

The section describing quantitative methods relies on the readers' familiarity with the WHO toolkit, which cannot be taken for granted. More detail would be welcome.

Authors' response:

We have now added the reference to our published work on analysis of national data using the same toolkit. And in the interest of word limit requirement by the journal, we just added one sentence on page 11, line 1,2 to describe the mentioned reference:

"Details of the toolkit were discussed in our previous similar work [11]"

Reviewer 2.4: As presented in the methods, three of the 4 (completeness, timeliness and accuracy) measure inter-level reporting processes, rather than quality per se: i.e. they measure what goes in and comes out, but not the extent to which the quality of what goes in (i.e. the records kept at facility level) accurately represents what happens at the facility. This is an occupational hazard for those trying to do retrospective data quality assessments, but given the prominence in the qualitative data of evidence of active mis-reporting, it at the very least merits some discussion. It would be especially interesting to know how evidence of mis-reporting is collected in sties that use standardised electronic data forms, since these are designed automatically to aggregate the data entered for reporting to the next level.

Authors' response:

Thank you for pointing this important methodological challenge with this data quality assessment approach. We agree that data can be complete, timely, accurate along the chain of reporting but wrong. We have actually seen this in our previous work at national level, where we compared the

routine health information data with survey and other gold standard sources for external consistency. At the local level we didn't do this, but we have added this in the limitation section (page 25, line 3, 4): "The data quality assessment tool we used may not rule out mistakes or wrong reporting in the RHIS processes."

Reviewer 2.5: Also, out of curiosity, how do you deal methodologically with the retrospective data completion described on p16 line6? Does the higher level get scored for over-reporting?

Authors' response:

You are probably right: what we call over-reporting is often taken as better performance and we indicated it as one of the problems in our discussion (page 25 line 13, 14)

Reviewer 2.6: Re qualitative data coding: the methods describe parallel coding and synthesis of a framework (p12 line29), which was then iteratively developed. What was the original framework based on? Since evaluations of HIS are not uncommon, it would be helpful to future researchers if the authors could share their final coding framework as supplementary material.

Authors' response:

Thank you for this comment. From the start we used PRISM framework to design the interview guide. During coding we didn't bind ourselves with any theoretical framework to allow and accommodate emerging themes. Later during analysis, we used the same PRISM framework to regroup the thematic area and present the result systematically. The final coding framework is now shared as supplementary material.

Reviewer 2.7: Results

Please provide the interviews in Table 1 by gender (and, for the fun of it, consider whether gender representation in service delivery may have some bearing on the variance of "quality" of reporting of different indicators reported mentioned on p16 line3)

Authors' response:

Thank you for your comment, we have now added gender for the qualitative interview Participants (Table 1) and 36% of the qualitative interview participants were female. Although gender could play a role in the quality of reporting, we don't have data to do such analysis. We agree it could be worth exploring the issue in future undertakings.

Reviewer 2.8: Re structure of the analysis: I understand that the authors are conducting their thematic analysis according to the PRISM framework. One limitation of that framework is that it does not sufficiently consider the role of incentive structures, both institutional and personal. Analysts thus struggle to classify important systemic drivers of poor performance (such as: "I'm already working a 12 hour day in a stressful situation to save lives for low pay, and now you want me to fill the same information into three spreadsheets to that you can report to different donors and maybe, if we've had a bad month or you've overcalculated the denominator, dock our budget for not meeting targets? No chance!") into 'organisational' or 'behavioural'. The paper rightly underlines the interplay between the PRISM categories (p9 line39); incentive structures are very often the thread that links them, and merit more explicit consideration, at least in the discussion.

Authors' response:

As you rightly mentioned, a limitation of the PRISM framework is its inclusion of the role of incentive structures but components are included under both organizational, and behavioural factors. We tried to frame our discussion differently from the way we presented the results because we felt the PRISM framework would not allow us to adequately highlight all our findings. We used the RHIS process to discuss matters arising at each stage, starting from Data processing to final data use. In doing so, we captured the role of incentives at different stages of RHIS process as effect of parallel reporting and use of multiple forms, at data processing, workload throughout the RHIS process and negative incentive mechanism at data reporting stages. However, as you suggested it is worth discussing the limitation of the RPISM framework based on our experience. Hopefully the readers will understand the complexity by reading the direct quotes from respondents. We have now also added a line in the Discussion under Limitations. (page 25, line number 4,5,6):

“It could be argued that PRISM framework used to guide our analysis may not clearly delineate some of the factors to either behavioural or organizational factor.”

Reviewer 2.9: A consideration of what the 117 interviews through the lens of incentives might also shed some light on the reasons why some of the most obvious and frequently-cited recommendations about health information systems have not actually been implemented in Ethiopia, despite the nominal importance of the Information Revolution in the government's health plan. Many of these are mentioned in the paper; they include such obvious and well-rehearsed elements as availability of reporting forms in appropriate languages, reduction of burden of reporting at the facility level, prioritisation of data for which there is local utility and demand, and ring-fenced funding for data systems.

Authors' response:

Thank you for this insightful analysis and we agree. In the current structure of the health system which is very hierarchical may not be possible to see the change that is initiated at a lower level. Most of the problems raised were coming from the lower level of the health system and may not have enough voice that could influence decisions at the higher level. Thus, this is one of the aims of the ORCA project that targets analysts within the health system, at the national or Federal level where big decisions are made. We hope for the ORCA participants to be a cohort of change agents from within. We added some more lines in the discussion to highlight the challenge with such a hierarchical structure and the potential of blame-shifting. Page 27 line 3-7:

“Respondents may have recognized elements of “blame culture” in the Ethiopian RHIS, described by others as emerging where hierarchical management structures reward compliance over efforts to expose poor quality and function [34]. The result is that staff eschew negative attention, which does not predispose them to raise awareness of systemic weaknesses or help develop genuine accountability.”